## Definitive evidence of the Mediterranean Outflow heterogeneity. Part 2: all along the Strait of Gibraltar

#### Claude Millot

Les Katikias, 83150 Bandol, France,
 *Correspondence to:* Claude Millot (ailesetiles@gmail.com)

Abstract. We have demonstrated in Part 1, with only a CTD transect across the Strait at 6°05'W, that the Mediterranean Outflow (MO) was definitely heterogeneous there. A yo-yo CTD time series has also provided

- astounding examples of both the marked layering that the Mediterranean Waters (MWs) display on the vertical at the Strait entrance (5°43'W), i.e. just upstream from the Camarinal sills (5°45'W), as well as the tremendous instability processes occurring in all layers. We focus herein on similar data collected within the Strait at both 5°50'W and 6°05'W (downstream from the Camarinal and Espartel sills, resp.) during five campaigns of the 1985-1986 GIBEX. We first show additional transects supporting the demonstration we made at 6°05'W, and we
- demonstrate that the marked heterogeneity of the MO within the Strait is clearly on the horizontal; as we expected, densest (resp. lightest) MWs flow on the bottom on its left-hand (resp. right-hand) side and all MWs are juxtaposed side by side. We also demonstrate that the density range within the MO in the western side of the Strait (6°05'W) is at least 0.5 kg.m<sup>-3</sup>, which is the density range, in the vicinity of the Cape St Vincent (8°30'W), of the four veins formed by the MO splitting. We show that the lightest component of the MO has started to be
- split as soon as Camarinal sills and sink all along the Strait. The splitting of the MO into veins is thus mainly due to its intrinsic heterogeneity, which is a direct consequence of the Sea functioning and of the mixing, within the Strait itself, of the MO with this or that type of Atlantic Waters (AWs). Therefore, the bathymetry in the Strait, and even in the Strait exit surroundings (near 6°20'W), has no major effect on the MO characteristics in the whole Ocean. We also focus on a yo-yo CTD time series collected during ~24 h at 6°05'W which shows that
- markedly different MWs have been passing by, clearly demonstrating that the horizontally heterogeneous MO is significantly meandering within the Strait. Finally, we confirm one of our previous results that, provided the temporal variabilities of both the MWs and the AWs are not too large, significant relationships can possibly be established between the characteristics of the MWs at the Strait extremities, or at least that the slope of the mixing lines on a q-S diagram provides significant information. Parts 1 and 2 of our trilogy must be assimilated before reading Part 3.

Keywords: Strait of Gibraltar, Mediterranean Outflow, heterogeneity, Mediterranean Waters

| 35 | Contents                                                                     |    |
|----|------------------------------------------------------------------------------|----|
|    | 1. Introduction                                                              | 3  |
|    | 2. The MO heterogeneity from cross-Strait transects                          | 6  |
| 40 | 2.1. General considerations                                                  | 6  |
|    | 2.2. The LYN and GIB Transects                                               | 8  |
|    | 2.2.1. At 6°05'W (Nov. 15, 1985)                                             | 8  |
|    | 2.2.2. At 5°50'W (Nov. 13-14, 1985)                                          | 10 |
|    | 2.2.3. At 5°50'W (Nov. 3, 1985)                                              | 13 |
| 45 | 2.2.4. At 6°05'W (Nov. 1-2, 1985)                                            | 15 |
|    | 2.2.5. At 5°50'W (Apr. 18-19, 1986)                                          | 17 |
|    | 2.2.6. At 6°05'W (Apr. 19, 1986)                                             | 19 |
|    | 2.2.7. At 5°50'W (Sep. 29, 1986)                                             | 21 |
|    | 2.2.8. At 6°05'W (Sep. 26, 1986)                                             | 23 |
| 50 | 2.2.9. At 5°50'W (Jun.19, 1986)                                              | 24 |
|    | 2.2.10. At 6°05'W (Jun. 18-19, 1986)                                         | 26 |
|    | 3. The MO heterogenity from the LYN3 yo-yo CTD time series at 35°50'N-6°05'W | 27 |
|    | 3.1 The three groups                                                         | 28 |
| 55 | 3.2 An attempt to associated MWs with CTD data at 6°05'W                     | 37 |
|    | 4. Discussion                                                                | 42 |
|    | 4.1 The overall density range of the MO                                      | 42 |
|    | 4.2 The long-term hydrological changes and the MO composition                | 42 |
| 60 | 4.3 The MO variability within the Strait                                     | 43 |
|    | 5. Conclusion                                                                | 44 |
|    | Acknowledgments                                                              | 46 |
| 65 | References                                                                   | 46 |

References

#### **1** Introduction

The Mediterranean Sea is a machine which, thanks to evaporation exceeding precipitation and river runoff over

- and all around the Sea, transforms waters from the Atlantic Ocean (AWs) inflowing at the surface, through the relatively narrow (~10 km) and shallow (~300 m) Strait of Gibraltar, into a set of intermediate (IWs) and deep (DWs) Mediterranean Waters (MWs). These MWs must exit from the Sea, hence forming the Mediterranean Outflow (MO) that has been shown for a while to be split into a series of veins, at least from the Strait exit. Because the MO is then identified in most of the northern Ocean, its splitting has received much attention but the
- MO has always been postulated to be relatively homogeneous upstream, if not from the Strait entrance (as in papers before the 2000's), at least within the Strait itself (as in 2015-2017 papers), the heterogeneity evidenced there being just considered as natural variability. And strangely, considering the dramatic importance and tremendous interest of the processes involved in the splitting, neither a single in situ experiment nor a single theoretical analysis have ever been dedicated to it. With our own papers about the functioning of the Sea, and in
- particular with a series of papers dedicated to the Strait (since Millot et al., 2006) in which we have been providing sound arguments about the possible identification of different MWs all along the Strait, we have thus generated an actual and major controversy about the homogeneity vs. heterogeneity of the MO.

We have not been convincing enough since a 2015 paper specifies being now "... in good agreement with the
previous study of Millot (2014b)", but also "While up to four MWs are spatially distinguishable east of the main
sill of Camarinal in the Strait, most of their differentiating characteristics are eroded after flowing over this
restrictive topography due to mixing. West of the sill, therefore, speaking of a unique Mediterranean Water
seems more appropriate" (Naranjo et al., 2015). And Garcia-Lafuente et al. (2017) clearly claim that, even
though IWs (resp. DWs) are found in the north (resp. south) of the Alboran just before the Camarinal sills (for

- reasons completely different from those we invoke), "the severe mixing and dissipation that takes place ... downstream ... blurs this spatial pattern and tends to form a rather mixed outflow ... in which the MWs are barely distinguishable". Consequently, it is not surprising that studies about the Strait exit, furthermore since mainly dealing with pure strait dynamics, assume a homogeneous MO there and within the Strait itself. Details are provided in the Background Sect. 2 of Part 1 but such an assertion is now clearly and definitively invalidated
- by the demonstration we made in Sect. 3 of Part 1 that we will support and complete herein.

We first propose (Sect. 2) additional examples of the MO heterogeneity within the Strait from five 1985-1986 GIBEX CTD cross-Strait transects at both 5°50'W and 6°05'W (Fig.1a): LYN1 (1-3 Nov. 1985), LYN2 (13-15 Nov. 1985), GIB1 (18-19 Apr. 1986), LYN3 (18-19 Jun. 1986) and GIB2 (26-29 Sept. 1986). We focus on the

100 deeper part of the  $\theta$ -S diagrams which all are relatively straight mixing lines within the Strait, what we evidenced in previous papers (e.g. Millot 2009, 2014a), and we emphasize the occurrence of relatively thick homogeneous layers lying on the bottom with markedly different  $\theta$ -S- $\sigma_q$  characteristics associated with different MWs that are thus roughly juxtaposed side by side. Now, focusing on the deeper parts of the  $\theta$ -S diagrams must be done without forgetting the tremendous variability induced by the occurrence, in either parts or the totality of

the Strait, of the two types of AWs that are the Surface Atlantic Water (SAW) and the North Atlantic Central Water (NACW), as shown in Fig.1b. Moreover, MWs can mix with either the base of SAW, as on 7-8 Nov.1985, or with its upper part, as on 13-14 Nov. 1985, that is just a week apart and nearly at the same location in the eastern side of the Strait. Note that the scale Δθ/ΔS in Fig.1b ((14.6-12.6)/(38.5-35.5) =2/3 °C) is different from the scale of all other figures below that is 1 °C, for both Sect. 2 ((13.6-12.8)/(38.55-37.75)=0.8/0.8 °C) and Sect.
3 ((13.14-12.94)/(38.32-38.12=0.2/0.2 °C).

Figure 1a. Bathymetry (m) of the studied area covering both the Strait of Gibraltar itself and its entrance, from 5°45'W (-5.75), in the western Alboran sub-basin. The five north-south CTD transects at ~5°50'W and 6°05'W
are indicated in dark blue. The two yo-yo CTD time series at 35°55'N-5°43'W (mainly analyzed in Part 1) and 35°50'N-6°05'W (analyzed herein) are located in yellow and displayed in Fig.1b. The moored CTD time series from HYDROCHANGES are located in green for both the southern Camarinal sill (5°45'W) and the southern continental shelf (5°43.5'W), and in light blue for the southern Espartel sill (5°59'W). The positions of the 1985-1986 GIBEX profiles (transects and yo-yo time series) were estimated from radar measurements.