# Peer review of "Definitive evidence of the Mediterranean Outflow heterogeneity. Part 2: all along the Strait of Gibraltar"

_Ocean Science, 2017_

## Referee Comment (RC1) · Anonymous Referee #1 · 21 Sep 2017

General comments The presence in the Gulf of Cadiz and further downstream, of four distinct layers of Mediterranean Water constituting well identified cores, has been generally attributed to bathymetric effects in the Gulf, although some authors sustain the existence of this heterogeneity already in the Strait of Gibraltar. The main reason for this controversy lies in the lack of appropriate historical data in the Strait itself. The present manuscript, which is the second part of a sequence of three, provides evidence of heterogeneities of the Mediterranean Outflow along the Strait and is based on a set of CTD transects and yo-yo time series within the Strait. The main objective is to show that the Mediterranean Outflow is already heterogeneous in the Strait itself. The importance of the present manuscript, which complements the first part relative to the Strait entrance, is not only the evidence of the heterogeneity of the outflow within the Strait

but also the demonstration of the spatial and temporal variability of the Mediterranean Outflow and the suggestions on the sampling strategy in such a complex area as the Strait of Gibraltar.

Specific comments In general, the written text (as happens with the first part of the series) could benefit from clarification of the text in some places. The figures illustrate the main conclusions of the manuscript, but some of them could improve by clarifying the respective captions. Page 10, Fig. 2b; page 13, Fig. 2c; page 24, Fig. 2i: the light gray lines of the yo-yo time series are almost invisible in a print Lines 183-188: make more clear the sentence "In addition, . . .Strait entrance" Lines 587-590: clarify the sentence contained in these lines Page 28, Fig. 3a caption: the meaning of the light blue lines and of the dark blue lines should be referred (with mention to the "first group" defined in the text) Page 29, Fig. 3b caption: refer to the "second group" Page 31, Fig. 3d caption: refer to the "third group" Lines 735: clarify the sentence "we inferred. . ." Lines 765-773: clarify the text, which is rather confusing

Technical corrections In the whole text, there are several cases of wrong letterings (normal instead of symbol) for the potential temperature (q instead of theta) and potential density anomaly (Sq instead of sigma-theta) Abstract, line 16: the meaning of "left-hand" or "right-hand" depends on the way you are looking at. Line 106: maybe Fig. 1 of Part I should be referred or, otherwise, give the theta,S values for SAW and NACW Line 109: figures below, which is 1oC,. . . Fig. 1a caption: Since latitudes and longitudes in the figure' axes are in decimal format, there should be a correspondence when lat. or long. values are referred in the caption, e.g., 5o 50'W (- 5.8 oW) Line 163: Millot (2008) is not in the References Fig. 1c caption: explain the meaning of the dot and the cross within circles Line 267: few hours apart Line 269-270: link the m with the exponent -3 (m-3) Line 439: one can retain Line 548: while in its upper part it is similar. . . Line 560: when referring to "the orange MW", a reference should be made to Table 1 of PartI Line 562: this light (?) Line 604: is close to the . . . Line 705: homogeneous, in cyan, . . .., in blue, . . . Pages 35, 36 and 37: Figures 5a, 5b and 5c should

all have the same depth intervals (either 25 m or 20 m) in the z scale Line 763: much more contrasted Line 798: ïĄşïĄś (orange) Line 1021: Millot (2008) is missing in the references (although it is referred in the text) Line 829: in particular the... Line 865: clarify "MO so as follow" Line 879: define "Mediterranean Inflow" Line 884: S displays Line 942: 1-2 days before Line 949: CTD through Line 986: Baringer & Price, 1997b is not referred in the text Line 990: CIESM group, 2001, is not referred in the text Line 994: García-Lafuente et al., 2011, is not referred in the text Line 1010: Millot, 2013, is not referred in the text

---

## Referee Comment (RC2) · Anonymous Referee #2 · 2 Jan 2018

the paper presents valuable data and evidence of heterogeneity of the MW current in the Strait of Gibraltar. But the paper is also very polemical and this does not have its place in a major journal. Referring to certain recent papers to support polemics is limited; many older papers do not claim homogeneity of the MW in the straits.

I am surprised that the papers by Madelain (1970) or by Zenk (1975) are not mentioned.

Furthermore, the paper goes all the other way, which is excessive. By declaring that the MW heterogeneity is sufficient to make it form several veins in the Gulf of Cadiz, the author denies the role of further diapycnal mixing on the gulf slope (clearly shown by Price and Baringer 1988 and later on by Cherubin 1997), or by the topographic steering effect of the canyons in the gulf. I strongly believe that all polemical aspects of this paper must go before it is published (part of them in the abstract, part of them in

the introduction, part of them in the conclusion).

Secondly, the paper offers little dynamical interpretation of the data. Assessing the role of bottom friction on the MW in the strait, of the entrainment of AW at the top of the MW layers, calculating orders of magnitudes of the diapycnic mixing rate in the straits, characterizing the mixing due to the internal waves, comparing the time for mixing with the time for advection (in a simple calculation I did, about 20 times longer) would give more support to the author's claim.

Thirdly, the paper contains many words expressing uncertainty "it is clear that, must be assumed, probably, might be, resembles, hypothesized..." or excess "tremendously, tremendous, dramatic consequences..." which are not quantitative and provide little information. They also must go.

typos and corrections abstract : and sinkS along the strait introduction : is a DYNAMICAL AND THERMODYNAMICAL machine which... figures 1a and 1b I cannot see the correspondence in geographical locations for the same colors lines 270-275 : the discrepancies seem dismissed here

all acronyms and variable names : $q(E)$, $S(E)$, $MLS(E,C)$, $S(C)$, $sigma\_q$, $S\_q$... must be defined.

In view of these remarks, this paper is inappropriate for publication in its present state and must be sufficiently revised to suppress polemics and to provide more scientific arguments connecting the various observations and quantifying the physics involved in this process.

---

## Author Comment (AC1) · 7 Jan 2018

**"Definitive evidence of the Mediterranean Outflow heterogeneity.
Part2: all along the Strait of Gibraltar"**

**Answer to Referee #1** (Comments received on 21 September 2017)

Dear Referee #1,

Let us first of all very sincerely thank you for your careful reading of our manuscript, your very helpful comments and your efficiency in rapidly providing us with your comments, which explains why we are now apologizing for answering with such a large delay; you certainly understood we were waiting for the report from Referee #2 that we received only 2 days ago.

Before answering your comments one by one, please let us specify that, even though you did not express any willingness to review the revised version, we will resubmit this Part2 in a markedly modified form. We will take into account not only your and the Referee #2's remarks and comments but also those we received from the Editor and the Referees of the Part1 paper. Essentially, we have proposed to the Editor a splitting of the former Part1/3 in two (which will make a tetralogy), with a new Part1/4 presenting an overview of the heterogeneity aspect and introducing the Parts2/4 to 4/4 that will focus on the entrance of the Strait, the Strait itself and the exit of the Strait. Two figures will be moved from Part2/3 to Part1/4: the diagram in Fig.1b will be enriched and presented as Fig.2-Part1/4 while the schema in Fig.1c will be presented as a complement to Fig.1-Part1/3 as Fig.3-Part1/4. We are personally convinced that this will markedly improve both the "Presentation Quality" and the "Scientific Quality", hopefully maintaining the "Scientific Significance" of the paper you reviewed.

Please, even though we perfectly understand you did not think necessary to review the revised version and do not want to engage yourself in another review, let us specify that we would appreciate any "friendly" (i.e. not official) comment (even just a few words!) that you could send us in a fully anonymous way with the help of the OS office, either on our answers below or on the new version we plan to submit before mid-April 2018. In any case, we warmly thank you for all what you did for us.

*General comments*
*The presence in the Gulf of Cadiz and further downstream, of four distinct layers of Mediterranean Water constituting well identified cores, has been generally attributed to bathymetric effects in the Gulf, although some authors sustain the existence of this heterogeneity already in the Strait of Gibraltar. The main reason for this controversy lies in the lack of appropriate historical data in the Strait itself.*
I consider you do not only share my own thinking but that it is also **your personal opinion**.

*The present manuscript, which is the second part of a sequence of three, provides evidence of heterogeneities of the Mediterranean Outflow along the Strait and is based on a set of CTD transects and yo-yo time series within the Strait. The main objective is to show that the Mediterranean Outflow is already heterogeneous in the Strait itself. The importance of the present manuscript, which complements the first part relative to the Strait entrance, is not only the evidence of the heterogeneity of the outflow within the Strait but also the demonstration of the spatial and temporal variability of the Mediterranean Outflow and the suggestions on the sampling strategy in such a complex area as the Strait of Gibraltar.*
This is an exact synthesis of the paper and **I very much appreciate your support.**

*Specific comments*
*In general, the written text (as happens with the first part of the series) could benefit from clarification of the text in some places.*
**I agree, I think I caught your overall comment in this respect and I hope the Editor will accept my proposition so as to have stand-alone and more focused papers.**

*The figures illustrate the main conclusions of the manuscript, but some of them could improve by clarifying the respective captions. Page 10, Fig. 2b; page 13, Fig. 2c; page 24, Fig. 2i: the light gray lines of the yo-yo time series are almost invisible in a print*
I will darken these gray lines.

*Lines 183-188: make more clear the sentence "In addition, : : :Strait entrance"*
The previous sentence indicates that θ-S diagrams are straight mixing lines between a given AW and a given MW that has already encountered mixing: because this modified MW cannot be compared with other MWs above (as can be done at the Strait entrance where the MWs are superimposed on the vertical) and not considering what is demonstrated after with the prolongation of the mixing lines towards the unmixed MWs, this given MW cannot be identified/colored. In addition: i) upstream from where a profile has been collected, numerous examples show that the specific MW it evidences can have been mixed with a different AW, ii) the CTD was not equipped with an acoustic device allowing to guarantee that it was lowered down to just a few m above the bottom, iii) files just indicate the bottom depth at the beginning of the profile, not the depth when the CTD was the closest to the bottom, iv) for a straight mixing line, the three parameters display similar variations with depth, v) while numerical values at the Strait entrance can be identified to this or that MW, this cannot be done after such a mixing has occurred. Therefore: maximum densities ($\sigma_{max}$) cannot be objectively quantified, classified and colored, as done in Part1. I will make the sentence more clear.

*Lines 587-590: clarify the sentence contained in these lines*
The lightest ($\sigma_{max}$ ~28.5 kg.m$^{-3}$) MW is evidenced ... by the longest transects at 5°50'W (Fig.2g, 2i) … at ~36.04°N (~36°02'N, see inserted position diagram) … will, most of the time, not be evidenced at 35°50'N (location too much south). I will clarify the sentence.

*Page 28, Fig. 3a caption: the meaning of the light blue lines and of the dark blue lines should be referred (with mention to the "first group" defined in the text). Page 29, Fig. 3b caption: refer to the "second group" Page 31, Fig. 3d caption: refer to the "third group"*
We agree and we suppose you are dealing with the gray (not blue) lines.

*Lines 735: clarify the sentence "we inferred: : :"*
HydroChanges CTDs were simultaneously moored at Espartel (E) and Camarinal (C) southern sills giving time series: θ(C), S(C), θ(E) and S(E). Mixing lines (over time, fitted) at both places display, with a time lag of ~8 h, similar slopes, hence giving another time series MixingLineSlope(C, E). If, for instance, you start with the set MixingLineSlope(C, E), S(C), θ(E) and S(E), you can compute a $\theta_{inferred}$(C) that you can compare with θ(C). All details are given and illustrated in Millot (2014a), essentially showing that mixing lines slopes provide a significant information representative of a relatively large along-stream domain. The same "technique" is used in Part2/3 to check whether the mixing lines at 6°05'W gives, for an S representative of the MWs a representative θ. I will clarify the sentence.

*Lines 765-773: clarify the text, which is rather confusing*
Let us reproduce Fig.6b here below.
The blue curve is from the $\sigma_{max}$ at 6°05'W: it does not evidence any group.

The orange curve is from the $\sigma_{max}$ inferred (what we did with $\theta_{inferred}(C)$ in Millot (2014a) can be applied to any parameter) at 5°43'W from the mixing line slopes at 6°05'W and for a salinity typical for the unmixed MWs there, S=38.44. We could thus deal with $\sigma_{max,inferred}$(5°43'W or 38.44) and the similar curve in green is for $\theta_{inferred}$(5°43'W or 38.44).

The orange curve can be colored according to the isopycnals (in particular 29.08 and 29.09) defined from the yo-yo time series at 5°43'W (in Part1). The two sets of red MWs before and after the occurrence of the pink (momentarily violet) MW display different homogeneities that could represent two types of red MW (one is thus colored in brown.

A major interest of such a technique is that, at 6°05'W for instance, **lowering the CTD more or less close to the bottom**, hence reaching larger or smaller $\sigma_{max}$ values **does not matter**: what is important is the slope of the linear fit to the deepest values.

[Figure]

*Technical corrections*
*In the whole text, there are several cases of wrong letterings (normal instead of symbol) for the potential temperature (q instead of theta) and potential density anomaly (Sq instead of sigma-theta).*
**I am sorry but I did not check enough the conversion of my docx files into pdf ones and I did not realize that errors occurred in converting the Symbol format only in the end of my files (in this paper after l. 289-302 only) and in a very strange way, for instance on l. 553 and not on l. 554! I will obviously check the totality of my files in the revised versions.** Your comment being

exactly the same as a comment from Referee#1-Part1, **I warmly thank you for having accepted such a huge work!**

*Abstract, line 16: the meaning of "left-hand" or "right-hand" depends on the way you are looking at.*
When you are driving a car, I suppose you do not have any problem in turning on the right (left) which is on your right-hand side (left-hand side). Seems to me the parallel is obvious. I am sorry but I always think in terms of dynamics: in the northern hemisphere, the Coriolis effect deflect any motion on the right … whatever the direction of the motion is. In other words, the E-W orientation of the Strait of Gibraltar has absolutely no consequences and features would be exactly the same with a MO flowing to the east or through a north-south strait. I thus prefer dealing, in some occasions, with "dynamical terms" instead of "geographical terms"!

*Line 106: maybe Fig. 1 of Part I should be referred or, otherwise, give the theta,S values for SAW and NACW*
I am sorry but, in l. 106, I specify "… as shown in Fig.1b" (of this Part2/3). Don't you think all the info you require is in this figure?

*Line 109: figures below, which is 1oC,*
You might be right but I am not sure

*Fig. 1a caption: Since latitudes and longitudes in the figure' axes are in decimal format, there should be a correspondence when lat. or long. values are referred in the caption, e.g., 5o 50'W (-5.8 oW)*
I agree. Just note that I am doing all figures by myself and I don't know how to write automatically values in the degree-minute format. Even though correspondences such as between 40' and 0.66° is almost straightforward, I will add the correspondences.

*Line 163: Millot (2008) is not in the References*
I will check

*Fig. 1c caption: explain the meaning of the dot and the cross within circles*
I am sorry, these are classical symbols I have always seen everywhere: to remember them, you have to think to a pencil. The point represents the lead (writing side): when you see it, the pencil is directed towards you.

*Line 267: few hours apart*
I will ask the Editor

*Line 269-270: link the m with the exponent -3 (m-3)*
Libre-Office is not very powerful, I hope the OS software will cope with this.

*Line 439: one can retain*
Yes

*Line 548: while in its upper part it is similar*
Yes

*Line 560: when referring to "the orange MW", a reference should be made to Table 1 of PartI*
Yes

*Line 562: this light (?)*
The density value is the one characteristic of the lightest of the MWs there

*Line 604: is close to the*
Yes

*Line 705: homogeneous,in cyan, : : :., in blue, : : :*
I will modify the writing

*Pages 35, 36 and 37: Figures 5a, 5b and 5c should all have the same depth intervals (either 25 m or 20 m) in the z scale*
Yes

*Line 763: much more contrasted*
Probably yes

*Line 798: ïAˎsˎïAˎs´ (orange)*
Sorry, comment not understood

*Line 1021: Millot (2008) is missing in the references (although it is referred in the text)*
Yes, already said.

*Line 829: in particular the*
Yes

*Line 865: clarify "MO so as follow"*
To identify a given MW somewhere, it is necessary to sample the whole MO upstream, up to the Sea where all pure MWs can be identified with a single profile (superimposed MWs). Otherwise, a modified MW can result from any pure MW mixing with either SAW or NACW.

*Line 879: define "Mediterranean Inflow"*
l. 880: the MI is the counterpart of the MO: the M Inflow is the inflow of AWs into the Sea.

*Line 884: S displays*
Yes

*Line 942: 1-2 days before*
Yes

*Line 949: CTD through*
Yes

*Line 986: Baringer & Price, 1997b is not referred in the text Line 990: CIESM group, 2001, is not referred in the text Line 994: García-Lafuente et al., 2011, is not referred in the text Line 1010: Millot, 2013, is not referred in the text*
Thanks, I will better check.

---

## Author Comment (AC2) · 7 Jan 2018

**"Definitive evidence of the Mediterranean Outflow heterogeneity.
Part 2: all along the Strait of Gibraltar"**

**Answer to Referee #2** (Comments received on 2 January 2018)
* * *
Dear Referee #2,

Your first overall comment:
*the paper presents valuable data and evidence of heterogeneity of the MW current in
the Strait of Gibraltar.*
First of all, I note you agree "***the paper presents*** *valuable data and* ***evidence of heterogeneity"!!!***

**This is for me, and should be for the Editor, a key-comment which clearly emphasizes the valuable scientific interest of my work ... at least in the Strait itself, which is the region essentially concerned by the last-published studies, hence necessarily both upstream and downstream!**

However, before answering your other comments which, I hope you understand, is done **essentially to the attention of the Editor**, I would like to specify a series of points:

1) As I said in answering the five precedent reviews I received (you are thus the last one ... which offers me the occasion to **provide the Editor with some kind of "overall answer"** before he takes his decision), I would appreciate receiving your own "reaction" to my answers, what you could do in a fully anonymous manner via the OS editorial office. I have not been able yet to access your Report (that specifies your overall appreciation and your willingness to review the revised version or not), but this cannot change my "plans".
This being said, the Editor is aware about the fact that, even though this answer is the last one I write, my own answers about the two Part3 reviews are presently checked by my co-author who comes back to work on January 10, so that I cannot hope posting them before at least a couple of weeks. Furthermore these two reviews are relatively positive and do not ask for marked changes, my willingness is to provide the Editor with my answers to your "relatively positive -your first comment- and relative negative -below" comments **as soon as possible. Indeed, your review led me to propose efficient solutions to improve the whole set of papers and I would like to give the Editor a time for reflection as long as possible.**

2) A general comment I already received from the Editor is that all papers in the series should be "stand-alone" ones, with the first introducing the whole series. I clearly understood and accepted this very valuable comment and I already proposed him to split the Part1/3 in two, hence submitting a tetralogy. As soon as I finish the writing of this answer, hence not waiting for the Editor decision, I will start the elaboration of Part1/4 with already clear ideas in mind. In particular, **two figures will be moved from Part2/3 to Part1/4**: the diagram in Fig.1b will be enriched and presented as Fig.2-Part1/4 while Fig.1c will be presented as a complement to Fig.1-Part1/3 as Fig.3-Part1/4. I am convinced that this will markedly improve the "Presentation Quality", the "Scientific Quality" and the "Scientific Significance" of the paper you reviewed.

3) Even thought I provided the Editor with a list (as required by OS) of five potential referees, with three of them having told me they would accept reviewing the series of three papers, I consider **I have been unlucky** in having had only one Referee (#1), out of those three, actually aware of all my work since having reviewed my three papers.

4) I answer your comments with my own "language", for instance I do not deal with "the MW heterogeneity" but with "the MO heterogeneity" … and I do not use the general naming of "Gulf of Cadiz", just because "strait dynamics", and the dynamics of the Strait of Gibraltar in particular, have nothing to do with "gulf dynamics", be it the Gulf of Cadiz that should be concerned, mainly if not only, by continental shelf (stricto sensu) phenomena.

Your second overall comment:
*But the paper is also very polemical and this does not have its place in a major journal.*
is similar to the one from Referee#2/Part1 and, I think, "unfair". You will certainly agree with me that we will not convince each others so that **I propose the Editor the following OVERALL SOLUTION**:

**The last paper published (in 2017)** about the Strait focuses on the Strait itself (this Part2) and, together with the previous published one (in 2015), it **clearly synthesizes the general "timeless" opinion that has scientifically motivated not only this series of papers, but most of the scientific interest I have always had for Gibraltar,** much before my first dedicated publication (Millot et al., 2006). Based on this:

1) Considering **this 2017 paper is an Ocean Science one**, I (propose to) will "plagiarize" (essentially copy-paste) the Introduction Chapter of this paper to get the Introduction Chapter of my Part1/4 dedicated to an overview of the homogeneity vs. heterogeneity question. I hope that you and the Editor will admit that **for Ocean Science in particular, this is an irrefutable and acceptable way to present the question and introduce the other papers without any polemics.**

2) Still with the major aim of avoiding any polemics, I hypothesize that exactly reproducing parts of published papers, without any additional comment, is both fully legal and fully neutral. Therefore, **I (propose to) will then just copy-paste without any additional comment**, at the end of this Introduction Chapter of Part1/4, **only three portions of already published sentences**, what I previously did in this Part 2 (l. 84-92) and in other parts as well:
i) "... in good agreement with the previous study of Millot (2014b)"
ii) "While up to four MWs are spatially distinguishable east of the main sill of Camarinal in the Strait, most of their differentiating characteristics are eroded after flowing over this restrictive topography due to mixing. West of the sill, therefore, speaking of a unique Mediterranean Water seems more appropriate"
iii) "...the severe mixing and dissipation that takes place …downstream … blurs this spatial pattern and tends to form a rather mixed outflow … in which the MWs are barely distinguishable".
I (propose to) will just link these three portions of sentences without any additional comment, **hence forming a given paragraph** at the end of this Introduction Chapter of Part1/4. **And I (propose to) will just reproduce this given paragraph as the unique paragraph of the Introduction Chapters of Part2/4, Part3/4 and Part4/4**, just specifying that a more complete Introduction is provided in Part1/4.**

**This will fully satisfy the Editor's justified requirement of having a series of "stand-alone papers", with the first introducing the whole series, … while avoiding any polemical aspect.**

*Referring to certain recent papers to support polemics is limited; many older papers do not claim homogeneity of the MW in the straits.*
You probably know that I have mainly worked within the Sea and you might have read that I do not consider myself as a specialist of the Strait. I just consider myself as a specialist of the Sea who is

interested in understanding where are going the MWs he has studied and followed all along their course in the Sea.

Therefore, I am sorry but the only work I know that "*do not claim homogeneity*" is by Howe.

**Please, could you provide me** (anonymously via the OS editorial office if you want) with additional references and, hopefully (I am retired and do not have any access to free libraries), **with the pdf's of such papers**?

*I am surprised that the papers by Madelain (1970) or by Zenk (1975) are not mentioned.*
I am personally surprised (furthermore I maintain I have been "unlucky") that you did not check the references cited in the Part1 paper as well as in what I (and the Editor) consider as a reference paper for my previous works, i.e. Millot (2014a) … in which you will see that there is even a Madelain (1967) that you should know!
I have had in mind Zenk (1975) but have been unable to retrieve it. In case you have a pdf … please forward it to me.
Whatever the case, you certainly know that Madelain is actually at the origin of the hypothesis of "a homogeneous MO split by bathymetry" that is still supported by the most recent 2015 and 2017 papers previously mentioned.

*Furthermore, the paper goes all the other way, which is excessive.*
Don't **ALL** (not only the previously cited one) **other papers go their "other" one way??????????**
**In case you know papers hypothesizing heterogeneity, please let me know and send pdf's!!!**

*By declaring that the MW heterogeneity is sufficient to make it form several veins in the Gulf of Cadiz, the author denies the role of further diapycnal mixing on the gulf slope (clearly shown by Price and Baringer 1988 and later on by Cherubin 1997), or by the topographic steering effect of the canyons in the gulf.*
**I am sorry but I must reject such an assertion**:

**1)** You did not just have a look at the Part1 and Part3 papers! In particular, for what concerns your focus on the Strait exit (Part3), I do not "*declare*", I think I "provide evidence", if not "demonstrate". **Please, just have a look at them and let me know if you still think I "*declare*".**

**2)** Please, just consider the "heterogeneity/homogeneity" (chose the term you want) evidenced from the cross-MO transect in Part3, in particular Fig.2a/Part3 … that you will find very similar to all the θ-S diagrams shown in the Part2 you reviewed.
Notice that this cross-MO transect was performed clearly upstream from any marked topographic feature.
Then, just have in mind what you "declare" as your first overall comment: "*the* Part2 *paper presents valuable data and evidence of heterogeneity of the MW current in the Strait of Gibraltar.*"
With this in mind, please could you finalize a similar sentence: "*the* Part3 *paper presents ____ data and ____ of heterogeneity of the MW current* at *the Strait* exit, upstream from any marked topographic feature.*"?*
**Do you continue thinking I "*declare*"?**

**3)** In Part3, we analyze data (only my co-author participated in the MO-2009 experiment) that were collected:
-only for some of them as general surveys upstream from ~6°36'W,
-for most of them, and the cross-MO transect in particular, in a central zone near 6°20'W … that is markedly upstream from any marked topographic feature.

**Please, have a look at Fig.1b and 18 in Part3, or provide me with the bathymetric map you want, and let me know which are the marked topographic features that could be considered as responsible for the "heterogeneity/homogeneity" at 6°20'W, hence upstream from there!**

**4)** I do not "*denies*"! It is not my point to refer to and comment papers dealing with the Iberian continental slope that is much further downstream from the area I an interested in. Note that this slope extends even out of the gulf … so diapycnal mixing of the veins has nothing to do with the Gulf of Cadiz itself: you should deal with an "alongslope dyapicnal mixing". The same remark can be made with the canyons that only have their upper part "in the gulf", at the outer edge of the continental shelf; and canyons are classical features of all continental slopes worldwide, hence having dramatic effects of the alongslope circulation worldwide. But, as far as I know, diapycnal mixing and canyons have never been invoked to explain hydrodynamical processes upstream from ~6°20'W hypothetically leading to the splitting. **Please, could you acknowledge or provide me with adequate references?**

**5) I am sorry to say that, with your "*clearly shown*", you "clearly declare"!** The authors you cite are "simulers", or "modelers" as you probably use to name them, what I refuse to do: most of the "simulers" I have encountered during more of 50 years tend to think that they are producing "models" that, therefore, have to be retrieved or respected by the whole community, including experimentalists like myself. I will never forget a remark of a colleague of mine (named Nadia) who told me that my twenty-five (1-year, 1-h) current time series (off Algeria) were more or less bullshit compared to the thousands of "data" (as she named her numbers) she got from her "models"!
With my own language, **colleagues working with computers are doing simulations of actual processes and are "simulers" and, as an "experimentalist", I imperatively need to work with them,** just to have my hypotheses checked, and hopefully validated, by numerical computations (simulations) or equations. Would you have had a look at Part1, you would have noticed that I am asking for convenient and dedicated simulations!
Now, with such references, you might be a simuler. But, as a Strait specialist, you might also be aware of dedicated in situ experiments that would have addressed the "homogeneity vs. heterogeneity question". **Please, could you let me know which in situ experiment has already been dedicated to this specific question?**

**6) As an overall answer to your comment that I reject**, let me specify that neither simulations nor data analyses, as much sophisticated they could be, can be considered as definitive. Yes, I do think that "*the* MO *heterogeneity* indicated by the data sets I am showing *is sufficient to make it form several veins",* which does not mean other effects such as topographic ones at the Strait exit or alongslope diapycnal mixing do not influence the splitting (downstream from where it initially occurred) and the final characteristics of the veins in the Ocean. **As clearly indicated by my title, it just seems to me that "there is definitive evidence of the MO heterogeneity from the Strait entrance to the Strait itself (there supported by your "*the paper presents valuable data and evidence of heterogeneity")* and to the Strait exit"**.

*I strongly believe that all polemical aspects of this paper must go before it is published (part of them in the abstract, part of them in the introduction, part of them in the conclusion).*
Even though you, as well as Referee #2 of Part1 and the Editor (influenced or not by both of you) see polemics in my writing where I just see my willingness to expose, in a way as clear as possible, the "homogeneous vs. heterogeneous controversy", let me specify that I will markedly modify my writing and, **just because I would first of all like to publish in Ocean Science, I will strictly respect the final Editor's recommendations**.

Even though I obviously have the same "*polemical writing*" all along my papers, I just checked, as examples of discussion between us, what could be the concerned sentences in this Part2 abstract, hence focusing on the splitting as you previously emphasized. I identified four sentences:

1) "We also demonstrate that the density range within the MO in the western side of the Strait (6°05'W) is at least 0.5 kg.m$^{-3}$, which is the density range, in the vicinity of the Cape St Vincent (8°30'W), of the four veins formed by the MO splitting." Is this an observation worth to be specified? If yes, do you think it is correctly written and, if not, how would you write it? Whatever the case, **is this an observation possibly supporting a "major effect" of the heterogeneity on the splitting**, hence a "minor effect" of both the alongslope diapycnal mixing and the topographic effects?

2) "We show that the lightest component of the MO has started to be split as soon as Camarinal sills and sink all along the Strait." Is this an observation worth to be specified? If yes, do you think it is correctly written and, if not, how would you write it? Whatever the case, **does this observation made markedly upstream from the Strait exit support a "major effect" of the heterogeneity on the final splitting in the Ocean**, hence support a "minor effect" of both the alongslope diapycnal mixing and the topographic effects that occur markedly downstream from the Strait exit?

3) "The splitting of the MO into veins is thus mainly due to its intrinsic heterogeneity, which is a direct consequence of the Sea functioning and of the mixing, within the Strait itself, of the MO with this or that type of Atlantic Waters (AWs)."
First, considering only the second part of the sentence (from "its intrinsic heterogeneity ..."), do you agree that "intrinsic heterogeneity" means (is understood as) "within the Strait itself", hence is clearly the focus of this Part2 paper? If you answer "yes", and even though you did not (at least carefully) read Part1, **do you agree the heterogeneity in the Strait itself that is evident to you (your first overall comment) only results from the Sea functioning and AWs-MWs mixing processes**?
Second, **do you think that the first part of the sentence "The splitting of the MO into veins is (thus) mainly due to its intrinsic heterogeneity" is not justified by the two previous sentences**? And **don't you think that the "mainly" is sufficient to let some place for the other processes** (the alongslope diapycnal mixing and the topographic effects); **if not, what could be for you an acceptable writing?**

4) "Therefore, the bathymetry in the Strait, and even in the Strait exit surroundings (near 6°20'W), has no major effect on the MO characteristics in the whole Ocean." In case you consider such a writing is polemical, **what could be for you an acceptable writing?**

*Secondly, the paper offers little dynamical interpretation of the data.*
I consider "*little*" is a bit more than "no". **So, please, could you specify what are the "*little*" dynamical interpretation of the data I provide in this Part2 paper**?
Whatever the case, and even though you never deeply read any of my other papers (be they already published or still submitted as parts of this series), I am sorry to say that I consider myself as an "honest and objective scientist". I never published "bla-bla/interpretation" papers. The "dynamical interpretations" I have offered in the past were i) based on reliable data sets and sound analyses, and ii) offered as schematic diagrams. In the past, I have published, in particular, schematic diagrams for the circulation of the surface, intermediate and deep waters, first in the western basin of the Sea, then in the whole Sea; and I have also published schematic diagrams for the structure of mesoscale eddies in the Algerian sub-basin (references available upon request). And finally, **for what concerns this specific series of papers, I have published schematic diagrams / dynamical interpretation for the whole MO from the Strait entrance downstream to the Iberian continental slope in Millot (2014a), clearly mentioned in Part1 and reproduced as Fig.19-Part3, as well as schematic diagrams / dynamical interpretation of the AWs-MWs mixing as**

**Fig.1-Part1 and Fig.1c-Part2. You might be right but, sorry, I am unable to provide more dynamical interpretation than that!!!**

*Assessing the role of bottom friction on the MW in the strait,*
I deal with bottom friction of MW**s**,

*of the entrainment of AW at the top of the MW layers,*
I do **not** deal with **entrainment** of AWs at the top of the MWs **veins: I deal with AWs-MWs mixing** (as inferred from CTD profiles, i.e. **without any dynamical information**, i.e. without any information about which of the two layers pulls along the other; is it an entrainment of AWs at the top of the MWs or an entrainment of the MWs at the base of the AWs, I do not have adequate data allowing me to specify???), and I do **not** deal with "*MW layers*" since I think/demonstrate/claim/declare (whatever the term you prefer) that the series of **MWs that are superimposed (or layered, i.e. on the vertical) at the Strait entrance (Part1) are juxtaposed side by side (i.e. on the horizontal, from the left-hand side of the MO to its right-hand side, i.e. from south to north) as veins in the Part2 that you reviewed, linked to a not-yet considered Coriolis effect that would be increased in the Camarinal sills surroundings (due to a necessary increase in velocity)**.

*calculating orders of magnitudes of the diapycnic mixing rate in the straits, characterizing the mixing due to the internal waves, comparing the time for mixing with the time for advection (in a simple calculation I did, about 20 times longer) would give more support to the author's claim.*
**Sorry but I am unable**:
- to calculate orders of magnitude(s) of the dapycnic (or diapycnal?) mixing rates in the strait(s),
- to characterize the mixing due to internal waves,
- to compare the times for mixing and advection,
- to control such even simple calculations.
**I am only able to** show data, evidence significant features … and motivate dedicated simulations … or additional data analyses. This is clearly specified at the ends of the Part1 (l. 1076-1095), **Part2 (l. 960-969)** and Part3 (l.1770-1790) papers.
**Therefore, why don't you take the opportunity of this series of papers to submit, together with my tetralogy (hopefully near mid April 2018), your simple calculations that would give either more support to what I am "claiming" or make you joining the group of the homogeneity + topographic effect partisans???**

This being said, **please could you let me know which kind of process could, in your mind, lead to the kind of profiles shown at 5°50'W and 6°05'W, that are essentially relatively straight mixing lines displaying relatively homogeneous waters in their lowest part?**

*Thirdly, the paper contains many words expressing uncertainty "it is clear that, must be assumed, probably, might be, resembles, hypothesized..."*
**And so what?** Would you prefer assertions? I am sorry to say I think that "**definitive peremptory sentences**" can only be used by simulers who fix themselves their own hypotheses and framework. It is then very easy to say that, considering this and this and this, then one can for sure guarantee that and that and that! **As an experimentalist trying to analyze data, I can never be definitive: I can just say that there is "definitive evidence" for this or that feature and express hypotheses for the reasons leading to such a feature!**

*or excess "tremendously, tremendous, dramatic consequences..." which are not quantitative and provide little information.*

When dealing with Sea in situ data in particular, "quantitative and accurate but unverifiable information can be considered as big, but it can be totally false". **Such words expressing excess for you, and maybe for simulers in general, just express the exaltation a Sea experimentalist can have after having evidenced features she/he never expected, furthermore when these features support the hypotheses she/he previously made!**

**Please, be sure that I do not want to compare myself with so "tremendously big scientists"**, but don't you think that Archimedes, when he discovered buoyancy, or Newton when he discovered gravity or Galileo when he discovered the Earth's rotation had reasons for being "excited" … hence for probably using what was considered as excessive words by the "reluctant" persons? Sorry but this is a writing you have to use to fight against skepticism. Let me confess that, when I was fighting wit colleagues to convince them that the IWs from the eastern basin were flowing along the European continental slope in the western basin and did not cross the Algerian sub-basin directly towards Gibraltar as generally believed at these times, I was thinking "And yet they turn!"...

*They also must go.*
I understand you ask me to remove such qualifiers. **Please, am I allowed to differentiate a major consequence from a minor one, or should I only deal with consequences without any qualifier?**

*typos and corrections abstract : and sinkS along the strait*
You can be right but I want to be sure: I want to say that "this component **has started to be split and has started to sink". What would be a correct (and "elegant") writing?**

*introduction : is a DYNAMICAL AND THERMODYNAMICAL machine which...*
I am not sure I clearly see the necessity for having used capital letters for what is, I think, only a suggestion from you. **In any case, I disagree**:

**1)** Any "machine", since you agree with me that the Sea is a "machine", needs "energy" to function and produce what it "has been built for". Energy is electricity for a coffee machine, coal for a steam locomotive or gasoline for a car. Energy for the Sea is, **ONLY** (at least from my point of view), the water balance between Evaporation (of the Sea) for the output and both Precipitation (over the Sea) and River runoff (from the land) for the input (leading to the famous **E-P-R budget**). **Therefore, and at least for me, the Sea is "just" an hydric machine!** Note that, following the Cambridge Dictionary, I avoid using "hydraulic" which means "operated by or involving pressure". **And, at least for me, the major consequence of the water balance is just the difference between the Sea-level and the Ocean-level, leading the Ocean to cascade (sic) into the Sea.**

**2)** More specifically: temperature in the Sea does not have any major direct effect, even though a climate much warmer over the Sea would lead, via thermal expansion, to reduce the level difference, hence the intensity of the cascading; the same could be said for atmospheric pressure since, for instance, a mean pressure much larger over the Sea would increase the level difference; dynamical constrains, such as for instance wind stress, can obviously modify the inflow from the Atlantic but only at relatively small time scales.

**3)** In addition to the water balance, and **to understand most of what is occurring at Gibraltar …** as well as most of the circulation within the Sea (with some knowledge about the thermal+hydric meteorological forcing), it is then **ONLY** necessary to consider the **Coriolis effect**. Indeed, it is only this effect that makes the IWs (intermediate MWs) outflowing (sic) on the right-hand side of the MO (on the European side of the Strait) and the DWs (the deep MWs) overflowing (sic) on the left-hand side of the MO (on the African side of the Strait).

To conclude about the machine, **I can just accept adding "hydric"**, which will somehow synthesize all what is said in the remainder of this first sentence of my Introduction.

*figures 1a and 1b I cannot see the correspondence in geographical locations for the same colors*
I am sorry but **there is no correspondence** in geographical locations for the same colors. As clearly indicated in both the captions and the figures:
- Fig.1b **displays** CTD profiles at i) 5°43'W (light blue), ii) 5°50'W (dark blue) and 6°05'W (green)
- Fig.1a **locates** CTD profiles and time series at i) 5°43'W and 6°05'W (yellow, as yo-yo time series), ii) 5°50'W and 6°05'W (dark blue, as transects) and iii) 5°45'W and 6°05'W (green and light blue, as yo-yo time series)
Note that:
- Fig.1a aims at showing **strategies** (CTD profiles distributed along transects in dark blue, CTD profiles yo-yoed at specific locations in yellow, and CTD time series near the bottom at specific locations in green/Camarinal longitude and light blue/Espartel longitude)
- Fig.1b aims at evidencing the CTD profiles **variability** over both time (light blue vs. dark blue) and space (blue vs. green).

*lines 270-275 : the discrepancies seem dismissed here*
If I understand well your succinct comment, **I am not dismissing anything** and just say that "one general feature is not retrieved on a single transect": for instance, and on the basis of any sports records, men are stronger and/or more rapid than women; but during a single competition, in some sports at least, a female champion can beat a male champion!

Let me know explain **my point about the AWs-MWs mixing in general**, and more especially when differentiating the lightest and densest components of these two types of waters and, please consider the drawing I made **especially for you** (I hope you will understand I did not polish my writing as I use to do for my papers):

[Figure]

- for the MWs, the lightest (the IWs, one in orange) are always (sic) in the north, so that the densest (the DWs, one in blue) are always in the south; this is a direct consequence of the Coriolis effect that has a "tremendous" (sorry for this "excess"!) importance for the circulation in both the Sea and the Strait.

- for the AWs, the lightest (SAW in cyan) is generally (sic) found over the MWs in the north because the densest (NACW in green) is generally (sic) found over the MWs in the south; this is a direct consequence of the fact that the SAW is a surface water that is found everywhere in the west of the Strait and, probably (sorry for this "uncertainty"), circulates sluggishly before entering in the Strait area, while NACW circulates markedly (it is always found alongslope in the west of the Strait, hence necessarily -this is not an "excess"- constrained by the Coriolis effect, which allows evidencing its circulation).

- Therefore, in general (sic), SAW mixes more with the IWs (in the north) and NACW mixes more with the DWs (in the south).

Now, let us consider the mixing rates (the dashed lines in the figure) and make some hypotheses:

1) In the most simple case, which appears not to be the most general one, let us hypothesize roughly similar mixing rates (dashed black lines). In such a simple case, the maximum density (the $\sigma_{max}$ in our text) of the MO will thus be a function of latitude with maximum values (i.e. the blue cross) in the south and minimum values (i.e. the orange cross) in the north. Note that, in such a case, the density difference between the two MWs is not markedly modified by the mixing, or the density range for the MWs is not modified along the Strait … which is not what is generally observed (the density range increases downstream)

.

2) In the most general case, the mixing between the lightest of the MWs with SAW is more intense (the violet cross) than the mixing of the densest of the MWs with NACW (the blue cross); this is the most general case because the densest of the MWs always tends to sink more than the others, hence becoming deeper and deeper and leading to a reduced mixing with NACW while the lightest of the MWs remains the closest to the AWs, SAW in particular. As in case 1), the maximum density of the MO will still be a function of latitude with maximum values (i.e. the blue cross) in the south and minimum values (i.e. the violet cross) in the north, but the density differences between the two MWs will be increased by the mixing, which is what is generally observed.

3) In the most abnormal case, the mixing rate in the south is much more important (leading to the red dashed line and cross) than the mixing in the north (the black dashed line and the orange cross), thus leading the MO densities in the south (the red cross) to be lower than in the north (the orange cross). This is the most abnormal case just because, as exposed in 2), there are no normal reasons for that; but, as demonstrated by the example shown in our text, this can "probably" occur under specific circumstances!

**Do you understand, and hopefully accept, my point of view?**

*all acronyms and variable names : q(E), S(E), MLS(E,C), S(C), sigma_q, S_q... must be defined.*
**I am sorry but I did not check enough the conversion of my docx files into pdf ones and I did not realize that errors occurred in converting the Symbol format only in the end of my files (in this paper after l. 289-302 only) and in a very strange way, for instance on l. 553 and not on l. 554! I will obviously check the totality of my files in the revised versions.**

Now, **I hope you have understood my computations** and have eventually had a look at the results presented in Millot (2014a).

*In view of these remarks, this paper is inappropriate for publication in its present state and must be sufficiently revised to suppress polemics and to provide more scientific arguments connecting the various observations and quantifying the physics involved in this process.*
When just considering your comments above, and obviously keeping in mind your first overall comment *"the paper presents valuable data and evidence of heterogeneity of the MW current in the Strait of Gibraltar",* **I think it should be normal that the Editor ask for "major revisions", what I have already planned to do** … at least with the aim to *"suppress polemics"*. However, as I have tried to explain above, and apart from improvements in the writing, **I think I will be unable (or not willing)** *to provide more scientific arguments connecting the various observations and quantifying the physics involved in this process.*

To conclude, some points recapped **for both you and the Editor**:
- I have not been lucky in having three different Referees #2: none of them has had an overview of my whole work.
- I have not been lucky for this Part 2 because you are probably more a theoretician than an experimentalist. Whatever the case, I am sorry to say that you did not comment very much on the data themselves and on the specific analyses I make.
**- And you did not answer a major question that I would like to reformulate** on the basis of the three sentences published in the 2015 and 2017 papers as i), ii) and iii) of 2) in page 2:

* the sentence i) just indicates that, roughly, everybody agrees on the heterogeneity at the Strait entrance I address in my Part1.
* sentences ii) and iii) essentially claim for homogeneity within the Strait itself (typically near 6°05'W) that I address in this Part2.

Therefore, my question: **When considering the data sets and analyses presented in this Part2, would you characterize the MO at 6°05'W as homogeneous or heterogeneous and, in case of disagreement with these 2015 and 2017 papers, how would you formulate it?**

I hope you will accept answering my questions, discussing my own comments and helping me in providing me with the information you would consider as worth to be specified, and I thank you in advance for your help.

---

## Editor Comment (EC1) · B.B. Barnier (Editor) · 13 Feb 2018

Dear Dr. Claude Millot,

Now that I have received two reviews of the Part-2 paper of the series that you submitted for publication to Ocean Science (reference OS-2017-53). You also provided answers to the referees' comments. The period of Open Discussion is now closed (since 2 February).

Although you indicated in your answer to my recommendations relative to the Part-1 paper that you have decided to re-handle your trilogy and to resubmit all your work to Ocean Science before mid-April 2018, it is necessary to finish the review process of this Part-2 paper. I therefore address my recommendations for the revision of this

paper.

Note that because of your decision to re-submit you whole work in a different form (4 papers), I do not ask you to respond point by point to my comments or recommendations, but to seriously consider them when re-structuring and re-writing your work.

I carefully read the referee's comments to your paper and the answers you provided. I also read the paper myself. I consider that the paper presents relevant evidence of the heterogeneity of the MW outflow in the Strait of Gibraltar, but also that it suffers from major presentation flaws of similar nature as the Part-1 paper that lead me to ask for a major revision.

Note that I shall provide you with my recommendations for Part-3 paper in a few days, and this will be the occasion to comment on the ensemble of the trilogy.

Both referees express a need for clarification, simplification and synthesis. I had myself a great difficulty to read throughout the paper and to identify its main scientific outcome. The description of the T,S profiles (Fig. 2 and Fig. 3) is very fastidious, especially because it is not streamlined and no effort is made to synthetize the analysis regarding the objectives of the paper. So many details are given in such long sentences, that it is difficult to relate them to the main conclusion of the paper that the MO is organized in as a set of different components juxtaposed side by side. In addition, the paper very often mentions results presented only in the Part-3 paper so we feel that we are reading the conclusion of this Part-3 and lose the purpose of the analysis of Part-2. This should be avoided in the revised version. The paper also presents extended considerations of already published results (especially in the discussion) that are not directly relevant for the focus of the paper. Note also that the frequent use of words like tremendous, dramatic, astounding, etc., often seems out of proportion with regard to the changes or results they qualify. The effect, as I feel it and as mentioned also by referee #2, makes more harm than good to the results you want to convey. You may consider this remark when revising the papers.

Referee #1 main concerns are on the text clarifications. I recommend that your take these remarks seriously in revising the paper.

Referee #2 is far more critic regarding the paper. I retain three main comments that you must consider while revising the paper. Referee #2 notices a polemical character in the paper and insists that this does not have its place in a major journal. The emphasis given to the homogeneity vs heterogeneity controversy is also qualified as excessive. I read several papers mentioned in this "controversy" to make my own opinion. I tend to share the opinion of referee #2. This controversy deserves only a minor importance and seems to me as being semantic rather than scientific, and to rely mainly on what is meant by heterogeneity. As I recommended for the revision of the Part-1 paper, you should be more moderate and less focused on the "controversy". I also agree with reviewer #2 that the assertion given in the paper that the heterogeneity of the MO and the mixing with the AW is sufficient to explain the splitting into several veins and to neglect the role possibly played by the bottom topography. The analysis presented here does not support such a statement, which to be acceptable should also provide dynamical evidence of the no-role played by the topography. I can mention here a study that shows that the mixing induced by local change in the topography has a major impact on the mixing of the MO (e.g. Nash et al., 2012, G.R.L., Vol 39). Also the topographic steering of the MO should certainly have consequence on the acceleration of the MO veins and therefore enhance the Coriolis effect. You correctly mention that the MW and the AW continuously flows with large velocities (e.g. lines 198-211) and that certainly influences the mixing. However, the discussion of mixing presented in the paper generally discuss the vertical mixing as an isolated process, the influence of the advection remaining qualitative when mentioned. The remark made by Reviewer #2 that the discussion should consider the difference in the diffusive and advective time-scales is quite pertinent. If as indicated by the referee the vertical mixing time scale is 20 time longer than the advective one, this could have an impact on your analysis. Baringer & Price (1997 already distinguished the local mixing processes and the mixing related to entrainment (Bulk mixing process). The discussion of mixing and entrainment should

attempt to be process oriented and quantitative. The study, which is essentially based on a visual interpretation of T, S diagrams, is hardly situated in a synoptic circulation context (no circulation map is shown that could link the various observation sites). The revised paper should attempt to enhance the links of the analysis with the circulation.

Allow me a personal comment. After reading several papers presenting studies of the MO in order to increase my scientific knowledge of the ocean dynamics of the Strait, I think that the word "definitive" used in the title is not pertinent. The reason is that the definition of heterogeneity/homogeneity is relative and will likely remain so (it depends on the magnifying glass used to look at it). For example, the study of Naranjo et al., (DSR 2015) pretend that west of the sill, speaking of a unique MW seems appropriate. You probably disagree with such a statement, but it is consistent with the data analysis performed in this study (which uses a cluster analysis as magnifying glass). A different approach, for example the one you propose, can yield a different vision. Nevertheless, the one proposed in Naranjo et al. is consistent with the analysis that was performed. This emphasizes the relativity of the notion of homogeneity of the MO. Note that the comment of the second referee of the Part-3 paper about the destruction of the "myth of a homogeneous MO" goes in the direction of my remark.

Here is a comment that concerns your answers to the referees. The tone of your answers surprised me, as it appears often personal and even conflictual, not in proportion to the critics or remarks made. I think that this is largely why referees did not answered to your comments, which at the end is damaging to the open discussion process. I expect a more moderate attitude in the following of this review process.

Just a last short comment to tease you on your strict attitude about the use of Names and Acronyms. I do not find very consistent your use of Northern Ocean for the North Atlantic when you are commonly using Surface Atlantic Waters (SAW) and North Atlantic Central Waters (NACW). Shouldn't you use instead NOSW for Northern Ocean Surface Waters and NOCW for Northern Ocean Central Waters? I do not see any reason for not giving the North Atlantic its common name. Use the names and acronyms

that you think appropriate but please let others the same possibility.